# Knowledge, attitudes, and barriers to HIV testing among youth in Kumba, Cameroon: A cross-sectional qualitative community-based focus group study

**Frederick Nchang Cho**[1,2,3]☉*, **Marie Clarie Fien Ndim**[1]☉*, **Diane Zinkeng Tongwa**[1], **Christabel Afor Tatah**[1], **Franklin Ngwesse Ngome**[1], **Eugine Mbuh Nyanjoh**[1‡], **Andrew N Tassang**[4,5‡]

1 RAPHA Higher Institute of Health and Biomedical Sciences, Kumba, Cameroon, 2 Cameroon Baptist Convention Health Services – HIV Free/Strengthening Public Health Laboratory Systems, Douala, Cameroon, 3 Infectious Disease Laboratory, Faculty of Health Sciences, University of Buea, Buea, Cameroon, 4 Department of Obstetrics and Gynaecology, University of Buea, Buea, Cameroon, 5 Atlantic Medical Foundation, Mutengene, Cameroon

☉ Frederick Nchang Cho and Marie Clarie Fien Ndim are joint first authors.
‡ Eugine Mbuh Nyanjoh and Andrew N Tassang are joint senior authors.
* nchang.cho@gmail.com (FNC); Marieclairendim@gmail.com (MCFN)

## Abstract

### Background

Human Immunodeficiency Virus (HIV) remains a major public health concern in sub-Saharan Africa. In Cameroon, young people are disproportionately affected but underrepresented in HIV testing statistics.

### Objective

To explore knowledge, attitudes, and behaviours related to HIV testing among youth in Kumba, Cameroon, and to identify barriers to inform community-based interventions.

### Methods

A cross-sectional qualitative study was conducted using nine focus group discussions (FGDs) with 75 youth (52 females and 23 males) aged 18 - 35 years across four quarters in the Kumba II municipality. Participants were purposively sampled to reflect diverse educational and occupational backgrounds. Data were thematically analysed using Braun and Clarke's framework with NVivo Version 14.

### Results

Participants demonstrated high awareness of HIV testing services (90.7%) and transmission via sexual contact (96.0%), though knowledge gaps remained regarding

**Data availability statement:** All relevant data are within the manuscript and its Supporting Information files.

**Funding:** The author(s) received no specific funding for this work.

**Competing interests:** The authors declare no competing interest.

**Abbreviations:** AIDS, Acquired Immunodeficiency Syndrome; FGD, Focus Group Discussion; HIV, Human Immunodeficiency Virus; KAP, Knowledge, Attitudes, Practices, NVivo, Non-numerical Unstructured Data Indexing, Searching and Theorising Software; PLWHIV, People Living with HIV; SRS, Simple Random Sampling; WHO, World Health Organisation.

non-sexual transmission and testing procedures. While 93.3% had previously undergone HIV testing, 57.3% reported stigma and 46.7% raised confidentiality concerns as ongoing barriers. Female participants feared being labelled as promiscuous, while males cited social norms that discourage help-seeking. Most participants supported school-based or youth-centred community testing, emphasising the need for privacy and youth-friendly environments. Key motivators for testing included the desire to know one's status (82.7%), symptom appearance (28.0%), and unprotected sex (17.3%).

## Conclusions

Despite strong awareness and high testing uptake, stigma and confidentiality concerns persist among youth in Kumba. To enhance HIV testing rates, community-based strategies should prioritise mobile clinics, peer outreach, and confidential youth-centred services. Strengthening education about HIV transmission and demystifying the testing process may further reduce barriers.

## Introduction

Human Immunodeficiency Virus (HIV) remains one of the most critical global health challenges, with over 38 million people living with the virus worldwide (World Health Organisation [1,2]). The virus attacks the body's immune system, specifically cluster of differentiation (CD4) T-lymphocyte cells, which are essential for the body's defence against infections. If left untreated, HIV progresses to the Acquired Immunodeficiency Syndrome (AIDS), which severely weakens the immune system and may lead to life-threatening infections and cancers (Centre for Disease Control and Prevention [3]). Despite the availability of effective treatments such as antiretroviral therapy (ART), which can suppress the virus to undetectable levels and allow individuals to live long, healthy lives, there is no cure for HIV. Consequently, prevention, early diagnosis, and prompt treatment are crucial in reducing the spread and impact of the virus [4,5]. Early diagnosis and testing play a pivotal role in the management and prevention of HIV. Routine HIV testing allows individuals to know their status and begin treatment early, which is vital for controlling the virus and preventing complications [6–8]. Moreover, early diagnosis enables individuals to adopt preventive measures, reducing the risk of transmitting HIV to others. Testing also helps identify those who may not exhibit symptoms but are still infectious, contributing to the broader public health goal of reducing HIV transmission rates [9,10]. Unfortunately, stigma, fear, and misinformation continue to deter many people from getting tested, hindering global efforts to control the HIV epidemic. Therefore, improving access to testing services and increasing awareness of their importance is essential for the successful management and prevention of HIV.

HIV remains a significant public health challenge in sub-Saharan Africa (SSA), where the region bears a disproportionate burden of the global HIV epidemic. An

estimated 80% of adolescents living with HIV reside in this region, with ongoing efforts focused on reducing transmission rates and improving care for those infected [7,11,12]. In Cameroon, the HIV prevalence among adults aged 15 - 49 is estimated at 2.7% in 2018, with the Southwest Region, where Kumba is located, reporting a prevalence rate ranging from 2.5% to 5% in recent years [13,14]. Community-based HIV testing is a crucial strategy to increase testing rates, particularly among youth. Traditional healthcare settings often present barriers such as stigma, lack of confidentiality, and inconvenient hours, deterring young people from seeking HIV testing [15,16]. By integrating HIV testing into community spaces like schools, youth clubs, and public health events, these barriers can be overcome, ensuring easier access to services. Additionally, HIV self-testing (HIVST) has shown promise in increasing testing uptake among adolescents, allowing them more privacy and control over their health decisions [16,17]. Community-based testing services help with early diagnosis, encourage prevention efforts, and reduce the stigma associated with HIV testing, thereby contributing to better health outcomes and reducing transmission rates [18,19].

Understanding community-based HIV testing behaviours, particularly among youth, is essential for improving HIV testing rates and advancing the global fight against HIV/AIDS. In SSA, youth are among the most vulnerable populations when it comes to HIV infection, yet they remain underrepresented in testing statistics [17]. This discrepancy highlights the need for targeted interventions that encourage early diagnosis and timely treatment, which are critical for managing HIV and preventing its spread. Research focusing on youth attitudes toward HIV testing provides valuable insights into the barriers they face, including stigma, lack of awareness, and fear of positive results [20,21]. Stigma surrounding HIV testing is particularly pronounced in many communities, leading to reluctance among young people to seek testing [22,23]. The fear of being labelled as 'promiscuous' or being ostracised from social networks can discourage youth from accessing testing services, thereby delaying diagnosis and contributing to ongoing transmission. Additionally, a lack of awareness about the importance of regular HIV testing and available services is another significant barrier [24]. Many young people do not fully understand the benefits of early HIV testing, such as knowing one's status and preventing further transmission [25,26].

Community-based HIV testing has emerged as a promising strategy for addressing these barriers, particularly by reducing stigma and increasing access to services in familiar, trusted environments [16,27,28]. Understanding how youth engage with community-based testing can inform more effective interventions tailored to this demographic. This research will provide critical insights into how to design strategies that address the unique challenges faced by young people in accessing HIV testing, ultimately contributing to better health outcomes and a reduction in HIV transmission.

The study has the following objectives:

1. To assess knowledge, attitudes, and behaviours related to HIV testing among youth in Kumba.

2. To identify barriers to HIV testing, including social, cultural, and psychological factors.

3. To explore community-based strategies for increasing HIV testing among youth.

## Methods

### Study design

This study employed a cross-sectional qualitative research design to explore the knowledge, attitudes, and behaviours related to HIV testing among youth in Kumba, a locality in Cameroon, from 17 February 2025 to 24 March 2025. The design enabled an in-depth understanding of the factors influencing HIV testing uptake, particularly barriers such as stigma, lack of awareness, and social and cultural factors.

### Study area

The study was conducted in the Kumba II municipality of Kumba. Kumba is the administrative capital of Meme Division, located 75 km north of the regional capital, Buea [29,30].

Kumba serves as a strategic commercial and transit hub, with a population estimated at 144,413 as of 2024 [31]. The town features vibrant market activity, anchored by the Kumba Main Market and Fiango Market, alongside smaller local markets such as Mbonge Road Market, Three Corners Market, and Barombi-Kang Market. The local economy is driven primarily by trade in agricultural commodities, particularly cocoa, the region's leading cash crop.

The community is ethnically diverse, with the indigenous Bafaw people constituting the traditional custodians of the land. However, Kumba has witnessed a steady influx of non-indigenous populations, including migrants from other regions of Cameroon and neighbouring Nigeria. This demographic diversity is especially pronounced among young people, many of whom are either in school, self-employed in informal trade, or unemployed [32].

Kumba II, the study's focal area, is characterised by high population density, limited availability of youth-friendly health infrastructure, and varying levels of HIV-related awareness. These contextual features, along with active community structures and a high rate of youth mobility, make it a critical location for understanding HIV testing behaviours and barriers among adolescents and young adults.

## Definition of key terms

**Youth**: Individuals aged 18–35 years, consistent with national guidelines.
**Active youth groups**: Organised groups of young people engaged in social, educational, or community activities, identifiable through community leaders or health workers.
**Focus group (FG)**: A structured small-group discussion guided by the research team to explore participants' knowledge, attitudes, and behaviours regarding HIV testing.

## Study population

The study targeted youth aged 18–35 years residing in Kumba, Cameroon. Participants were selected from diverse backgrounds, including students, business people, and other working youth, ensuring comprehensive representation of the local population.

## Eligibility criteria

Participants were eligible if they were aged 18 years or older, had resided in Kumba for at least six months, and were members of a selected Focus Group (FG). They were also required to be willing to freely participate in the study.

Individuals outside the 18–35 age range, as well as visitors, passers-by, or non-residents of Kumba, Cameroon, were excluded to maintain the study's focus on the local community dynamics. Participation was entirely voluntary, and participants were free to withdraw from the study at any time.

## Sampling method

Multistage sampling was employed to systematically select participants across multiple levels; municipality, quarters, streets, and youth groups, ensuring diverse representation while accommodating practical constraints in accessing the population [30,33,34].

*First stage:* The Kumba II municipality was randomly selected from the three existing municipalities in the city of Kumba (Kumba I, Kumba II, and Kumba III) using simple random sampling (SRS).

*Second stage:* Four quarters within Kumba II; Kumba Town, Hausa Quarter, Kossala B, and Fiango, were selected using SRS from a pool of ten eligible quarters. Although six quarters were initially targeted, data collection was ultimately limited to four due to logistical and contextual constraints.

*Third stage:* Specific streets within the selected quarters were then randomly chosen to guide recruitment efforts. The selection was based on local knowledge, ensuring coverage of diverse residential areas.

*Final stage:* Nine focus groups, each comprising between 7 and 11 participants [35,36], were selected through purposive and convenient sampling to include, a mix of students, business persons, and other working youth. Recruitment was facilitated by local community leaders and health workers, who helped the research team identify and access existing youth groups within the selected quarters.

Recruitment was facilitated through collaboration with community leaders and local community health workers, who assisted in identifying active youth groups and linking the research team with focus group leaders and potential participants. This community-based approach helped ensure cultural sensitivity, trust-building, and inclusion of diverse voices across the selected quarters.

Data saturation, defined as the point when no new themes, ideas, or perspectives emerged from additional discussions, was used to determine the adequacy of the sample size. The research team conducted ongoing thematic reviews after each FGD, comparing emergent codes and topics. By the ninth FGD, subsequent sessions were yielding repetitive information, with no new insights identified during preliminary analysis. This confirmed that data saturation had been achieved, and further data collection was unnecessary [35].

Prior to participation, both verbal and written information about the study's purpose, procedures, and confidentiality measures was provided to all prospective participants. Written informed consent was obtained from each participant in accordance with ethical research guidelines. Ethical clearance for the study was granted by the Regional Delegate of Public Health for the Southwest Region of Cameroon.

## Data instruments and data collection

Data were collected through Focus Group Discussions (FGDs), facilitated by two field researchers. One acted as the facilitator, while the other served as a note-taker; with roles alternating across sessions. The HIV clinic staff, acted as co-facilitators, providing expertise to ensure accurate HIV-related discussion. The FGDs were conducted in neutral community spaces, including youth centres and meeting halls, with privacy ensured. The FGDs which provided an interactive and in-depth exploration of participants' knowledge, attitudes, and experiences regarding HIV testing (Fig 1; S1 Appendix). The discussions were semi-structured, guided by a set of pre-designed questions assessing HIV-related knowledge, testing behaviours, perceived barriers, and community-based strategies for increasing HIV testing uptake [37].

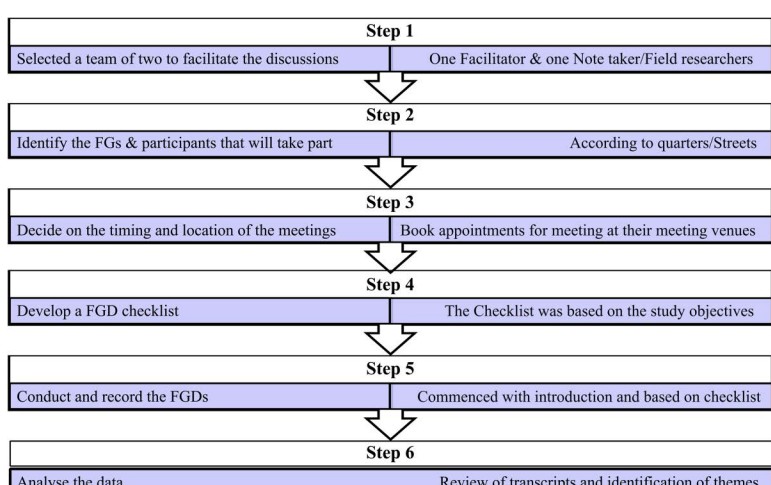

**Fig 1. Stages in the FGD process.** Sources: Hennink & Kaiser. (2022); INTRAC. (2017) [35,38].

Each focus group discussion (FGD) comprised between seven and eleven participants, lasting between 115 and 165 minutes (mean = 138 minutes). On average, each FGD included about eight participants (mean = 8). A total of nine FGDs were conducted, yielding rich qualitative data for thematic analysis.

## Ethical considerations

This study was conducted in strict conformity with the Helsinki Declaration [39] and received ethical approval from the Regional Delegation of Public Health for the Southwest Region, Cameroon (Reference number: P42/MINSANTE/SWR/RDPH/CB.PT/529/430). Verbal administrative clearances were obtained from Quarter Heads in all selected communities. Written informed consent was obtained from each participant in accordance with ethical research guidelines and institutional guidelines. Participants were informed of their right to withdraw at any point without consequence of loss of service, and that all information provided would be treated with strict confidentiality and used exclusively for research purposes. In cases where participants were under 21 (the age of majority in Cameroon), assent was obtained along with parental or guardian consent in line with national research ethics guidelines.

To ensure confidentiality during the focus group discussions (FGDs), participants were assigned unique identification numbers rather than using personal identifiers.

## Data analyses

Thematic analysis was employed to analyse the data, following Braun and Clarke's framework [40]. The research team transcribed all focus group discussions verbatim and systematically coded the transcripts using NVivo version 14 software (QSR International) [41]. Codes were then reviewed and grouped into overarching themes aligned with the study's objectives. This inductive approach enabled the identification of recurring patterns and unique insights into factors influencing HIV testing behaviours among youth in Kumba. The themes that emerged, knowledge, attitudes, testing practices, stigma, and community engagement, provided a nuanced understanding of both barriers and facilitators to HIV testing uptake within the local context.

## Results

This section presents findings from an inductive thematic analysis of nine focus group discussions (FGDs) involving 75 youth participants (69.3% female, 30.7% male) conducted across selected communities in Kumba, Cameroon. Thematic categories align with the study objectives and are organised into six core themes (Table 1). These reflect participants' knowledge, lived experiences, attitudes, practices, and suggestions regarding HIV and access to testing services.

Percentages presented indicate the frequency of coded responses within and across focus groups. They represent the relative emphasis of particular views rather than statistical generalisations.

## Sociodemographic characteristics of FGD participants

Participants were recruited from six neighbourhoods in Kumba (Hausa Quarter, Kossala B, Ntoko Street, Fiango Farm Road, Fiango New Layout 1, and Fiango New Layout 2). However, due to contextual and security-related challenges, data collection was successfully completed in four of these sites. These communities represent the city's main residential and social clusters, ensuring diverse youth perspectives. A total of 75 individuals participated in the FGDs. Their sociodemographic characteristics, including gender, education level, occupation, marital status, and religious affiliation are summarised in Table 2. This reflects a largely young, single, and educated sample with strong representation of students.

Most participants had completed secondary education (78.7%), and the majority were students (73.3%). The sample predominantly comprised Christians (92.0%), with a high proportion of participants reporting being single (84.0%).

**Table 1. Summary of key themes and subthemes.**

| S/N | Theme | Subthemes | Key Findings/ Illustrative Points | Reference |
|---|---|---|---|---|
| **1.** | Knowledge and Awareness of HIV and Testing Services | - Understanding of HIV | - 90.7% aware of HIV | Table 3 |
| | | - Modes of transmission | - 96.0% cited sexual transmission | |
| | | - Difference between HIV and AIDS | - 53.3% correctly identified HIV/AIDS distinction | |
| | | - Awareness and understanding of testing services | - 90.7% aware of testing | |
| | | | - 62.7% described procedures accurately | |
| **2.** | Experiences and Attitudes Towards HIV Testing | - Emotional reactions | - 37.3% felt very comfortable, 45.3% felt comfortable | Table 4 |
| | | - Peer encouragement | - 89.3% would recommend testing | |
| | | - Testing experience | - 69.3% reported positive testing experiences | |
| **3.** | Practices Towards HIV Testing | - Motivations for testing | - 82.7% Knowing one's HIV status | Table 5 |
| | | - Frequency | - 94.7% recommended quarterly testing | |
| | | - Influential factors | - 28.0% Symptoms appear | |
| | | - Support systems | - Health personnel (73.3%) most influential | |
| **4.** | Stigma and Barriers | - Fear of outcome | - 81.3% feared testing positive | Fig 2 |
| | | - Stigma and labelling | - 57.3% cited stigma | |
| | | - Confidentiality issues | - 46.7% concerned about confidentiality | |
| **5.** | Awareness Campaigns and Community Programmes | - Exposure and effectiveness | - 90.7% aware of community campaigns | Table 6 |
| | | | - 69.3% found them community campaigns effective | |
| | | - Suggested improvements | - 41.3% Preferred: mobile clinics, peer outreach, privacy-focused options | Table 6 |

Summary of participants' knowledge, experiences, practices, and perceived barriers regarding HIV testing, with key findings linked to supporting tables and figures.

**Table 2. Sociodemographic characteristics of FGD participants (n = 75).**

| Characteristic | Subclass | Frequency | Percentage |
|---|---|---|---|
| Gender | Female | 52 | 69.3 |
| | Male | 19 | 30.7 |
| Educational Status | Primary | 7 | 9.3 |
| | Secondary | 59 | 78.7 |
| | Tertiary | 9 | 12.0 |
| Employment Status | Student | 55 | 73.3 |
| | Business persons | 12 | 16.0 |
| | Farmers | 5 | 6.7 |
| | Teachers | 2 | 2.7 |
| | Electrician | 1 | 1.3 |
| Marital Status | Single | 63 | 84.0 |
| | Married | 12 | 16.0 |
| Religion | Christian | 69 | 92.0 |
| | Other | 2 | 2.7 |
| | Muslim | 3 | 4.0 |

FGD: Focus Group Discussion; Percentages may not total 100% due to rounding

### Knowledge and awareness of HIV and testing services

**Understanding of HIV and modes of transmission.** Most participants demonstrated a good understanding of HIV transmission routes. Most (96.0%) correctly identified sexual intercourse as a primary transmission route. Awareness of mother-to-child transmission (46.7%) and transmission via blood transfusion (30.0%) was moderate. However, some misconceptions persisted. A few participants (2.7%) believed HIV could be contracted through sharing food with infected individuals, and 1.3% cited intravenous drug abuse.

"*Many youths know about HIV, but they don't really understand everything about the testing process.*" (FGD5–Male)

"*HIV comes from sex without protection. Some people get it from blood or from the mother when she gives birth.*" (FGD2–Female)

Regarding prevention, 65.3% cited condom use, and 58.7% mentioned abstinence and fidelity. Only 1.3% stated that they were unaware of any prevention methods (Table 3).

Beyond general awareness, participants discussed how they distinguish HIV from AIDS.

**Understanding the Difference between HIV and AIDS.** Just over half (53.3%) understood that HIV can be treated while AIDS cannot. Nonetheless, 24.0% expressed uncertainty, 9.3% believed there was no distinction between the two, and 4.0% incorrectly assumed that both could be cured (Table 3).

**Table 3. Knowledge and awareness of HIV and testing services (n = 75).**

| Characteristic | Subclass | Frequency | Percentage |
|---|---|---|---|
| HIV Prevention Methods Known | Use of contraceptives | 49 | 65.3 |
| | Abstinence & Fidelity | 44 | 58.7 |
| | I don't know | 01 | 1.3 |
| Modes of HIV Transmission Known | Through sex | 72 | 96.0 |
| | Blood transfusion | 30 | 30.0 |
| | Mother-to-child | 35 | 46.7 |
| | Eating with HIV-positive individuals | 2 | 2.7 |
| | Intravenous drug abuse | 1 | 1.3 |
| Difference Between HIV and AIDS | HIV can be treated, but AIDS cannot | 40 | 53.3 |
| | I don't know | 18 | 24.0 |
| | There is no difference | 7 | 9.3 |
| | Both can be treated or cured | 3 | 4.0 |
| Aware of testing services | Yes | 68 | 90.7 |
| | No | 5 | 6.7 |
| | I don't know | 2 | 2.7 |
| Description of HIV test procedure | Correct sample and procedure | 47 | 62.7 |
| | Wrong procedure | 21 | 28.0 |
| | Correct sample and wrong procedure | 2 | 2.7 |
| | I don't know | 2 | 2.7 |
| Perceived benefits | Knowing one's HIV status | 62 | 82.7 |
| | Reduce community HIV transmission | 17 | 22.7 |
| | I don't know | 7 | 9.3 |

Participants were allowed to provide multiple responses per question; therefore, frequencies may exceed the total sample size (n = 75). Percentages are based on the number of participants.

"*Some of us think once you get HIV, that's the end. We don't know that treatment can help you live normally.*" (FGD4–Female)

"*People confuse HIV and AIDS; we need more education about the difference.*" (FGD7–Male)

**Awareness and description of testing services.** Awareness of HIV testing services was high, with 90.7% of participants indicating familiarity with local testing services. However, 6.7% were unaware, and 2.7% were uncertain. When asked to describe the testing process, 62.7% accurately explained both the sample collection and procedure, though 28.0% described the procedure inaccurately, and others were unsure or partially correct.

"*Some people just know there's a place for testing, but they don't know how it's done.*" (FGD6–Female)

**Perceived benefits of HIV testing.** Most participants (82.7%) viewed knowledge of one's HIV status as the main benefit of testing, while 22.7% highlighted its role in reducing community transmission. A minority (9.3%) were unaware of any specific benefits.

### Experience and attitudes towards HIV testing

**Emotional responses and peer encouragement.** Most participants expressed comfort with HIV testing; 45.3% felt comfortable, and 37.3% felt very comfortable (Table 4). However, 17.3% reported feeling uncomfortable. Additionally, 89.3% indicated they would encourage others to get tested, while 10.7% would not, largely due to fear of positive results and stigma (75.0%) or confidentiality concerns (25.0%) (Table 4).

"*When I tested, I was scared at first, but the counsellor made me relax.*" (FGD3–Female)

"*We tell our friends to go for testing because knowing is better than guessing.*" (FGD6–Male)

Beyond individual comfort, participants reflected on how community attitudes shape testing behaviour.

**Community attitudes and social labelling.** A majority of respondents (69.3%) perceived community attitudes towards HIV testing as positive. Nonetheless, 30.7% described their communities as unsupportive and stigmatising (Table 4). Common expressions included "they don't get treated well" and "people are afraid of being labelled".

"*In our community, people think if a girl goes for testing, she is wayward.*" (FGD8–Female)

**Table 4. Attitudes and perceptions of HIV testing among youth.**

| Characteristic | Subclass | Frequency | Percentage |
|---|---|---|---|
| Comfort Level with HIV Testing | Very Comfortable | 28 | 37.3 |
| | Comfortable | 34 | 45.3 |
| | Uncomfortable | 13 | 17.3 |
| Would Encourage Others to Get Tested for HIV? | Yes | 67 | 89.3 |
| | No | 8 | 10.7 |
| If no, why? | Fear of positive results and stigma | 6 | 75.0 |
| | Confidentiality | 2 | 25.0 |
| Community attitude towards HIV testing | Positive | 52 | 69.3 |
| | Poor | 23 | 30.7 |
| Ever Tested for HIV | Yes | 70 | 93.3 |
| | No | 5 | 6.7 |

*"Boys fear to test because they don't want others to think they are sick or careless."* (FGD9–Male)

**Personal experiences with testing.** Personal uptake of testing was high, with 93.3% of participants reporting having undergone HIV testing. Only 6.7% had never been tested, reflecting strong engagement with available health services (Table 4). The high testing rate may reflect both personal health awareness and the positive peer and community attitudes documented above.

## Practices towards HIV testing

**Motivations for HIV testing.** Across all FGDs, the overwhelming majority of respondents (82.7%) identified the desire to know their HIV status as the principal motivation for testing. A smaller proportion (22.7%) cited the aim to reduce HIV transmission within the community, highlighting an awareness of public health implications (Table 5). A minority (9.3%) expressed uncertainty or lacked a clear rationale, indicating gaps in knowledge or engagement.

*"I test every three months because it helps me stay safe."* (FGD4–Female)

*"If you have unprotected sex, it's better to test and be sure."* (FGD7–Male)

**Frequency of HIV Testing.** All nine FGDs indicated that routine testing was generally advised or practiced, with most participants (94.7%) recommending testing *every three months*, *following unprotected sexual encounters*, or when *symptoms appear* (Table 5). This consistency suggests a shared understanding of HIV testing as a periodic and conditionally triggered health practice. Only a small subset (5.3%) were uncertain about the appropriate frequency.

Participants also discussed how often one should test and under what conditions.

**Influential Factors.** Key factors influencing the decision to test for HIV were uniformly discussed across all FGDs. Participants reported a variety of factors that influenced their decision to seek HIV testing. The most common was the *appearance of symptoms* (28.0%), followed by concerns about *accessibility of testing services* (20.0%) and *confidentiality*

**Table 5. Self-reported practices towards and motivations for HIV testing among youth.**

| Characteristic | Subclass | Frequency | Percentage |
|---|---|---|---|
| Motivations for HIV testing/ Benefits Table 3 | Knowing one's HIV status | 62 | 82.7 |
| | Reduce community HIV transmission | 17 | 22.7 |
| | I don't know | 7 | 9.3 |
| Frequency of HIV testing | Three months | 71 | 94.7 |
| | I don't know | 4 | 5.3 |
| Influential factors | Unprotected sex | 13 | 17.3 |
| | Confidentiality | 14 | 18.7 |
| | Symptoms appear | 21 | 28.0 |
| | Stigma and fear | 2 | 2.7 |
| | Accessibility | 15 | 20.0 |
| Presence of support systems | Yes | 74 | 98.7 |
| | No | 1 | 1.3 |
| Support systems | Healthcare providers | 55 | 73.3 |
| | Testing counsellors | 3 | 4.0 |
| | Relationships (friends or parents) | 17 | 22.7 |

Frequencies may exceed the total sample size (n = 75) as participants could provide multiple responses, with percentages reflecting the proportion reporting each response

(18.7%). *Unprotected sex* was also a significant motivator (17.3%), though relatively few participants (2.7%) explicitly cited *stigma and fear* as barriers (Table 5).

**Support systems.** Participants across all FGDs emphasised the importance of social and institutional support in promoting HIV testing. Almost all respondents (98.7%) indicated the existence of *support systems* that could facilitate HIV testing. Among these, *healthcare providers* were most commonly identified as sources of support (73.3%). Others mentioned included *friends or parents* (22.7%) and *testing counsellors* (4.0%) (Table 5).

### Perceptions of stigma and Barriers to HIV testing

**Fear, misconceptions, and social judgement.** Participants identified several factors discouraging HIV testing as presented in Fig 2.

Female participants noted that being tested could be interpreted as evidence of promiscuity, while male participants discussed gender norms that discourage health-seeking behaviour, such as the perception that "boys fear being seen as weak".

*"People talk too much; they will say you are positive just because you went for testing."* (FGD5–Female)

*"Some of us hide because we don't trust the confidentiality at hospitals."* (FGD3–Male)

### Awareness campaigns and community programmes

**Participation and perceived effectiveness.** Nearly all participants (90.7%) reported being aware of HIV campaigns or community testing initiatives, and 86.7% had either participated in or observed them. The majority (69.3%) viewed these programmes as very effective, particularly when offered in schools or community venues (Table 6). However, 18.7% raised concerns about confidentiality and the potential for involuntary disclosure, particularly in school-based testing (Table 5).

*"When they bring testing to schools or markets, many youths go. But if it is at the hospital, people hide."* (FGD2–Female)

*"The outreach is good, but privacy must be respected. That's when more boys will come."* (FGD6–Male)

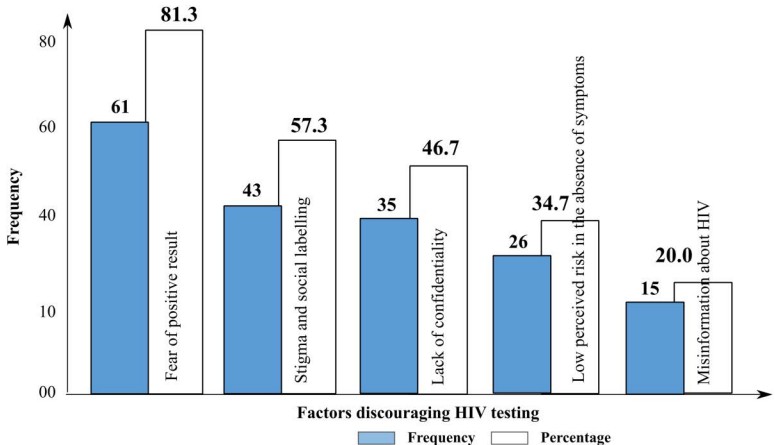

**Fig 2. Factors discouraging HIV testing.**

**Table 6. Effectiveness of HIV testing campaigns and community perceptions.**

| Characteristic | Subclass | Frequency | Percentage |
|---|---|---|---|
| Awareness of HIV Testing Campaigns in Kumba | Yes | 68 | 90.7 |
| | No | 7 | 9.3 |
| Participation in HIV Testing Campaigns in Kumba | Yes | 65 | 86.7 |
| | No | 10 | 13.3 |
| Perceived Effectiveness of HIV Testing Campaigns | Very Effective/Encouraging | 52 | 69.3 |
| | Effective/Encouraging | 12 | 16.0 |
| | Not Effective/Encouraging | 2 | 2.7 |
| | I don't know | 6 | 8.0 |
| Preferred Methods to Encourage Testing | Creating Awareness | 44 | 58.7 |
| | Bring services closer to youths | 2 | 2.7 |
| | Render accessible and affordable services | 16 | 21.3 |
| | Advocacy to the people | 13 | 17.3 |

Participants' awareness, participation, perceived effectiveness, and preferred strategies for HIV testing campaigns in Kumba

**Recommendations for improvement.** Participants proposed a range of improvements to existing awareness and testing efforts, as presented in Fig 3.

Frequency and percentage of participants' suggested improvements to existing HIV awareness and testing efforts Emphasis was consistently placed on non-judgemental, private, and youth-friendly environments.

## Discussions, recommendations and conclusions

### Interpretation of key findings

This study investigated the knowledge, attitudes, and practices regarding HIV testing among youth in Kumba, Cameroon. The findings reveal key insights into the factors influencing HIV testing uptake, including levels of awareness, social perceptions, and community-based strategies for increasing testing. The study's results demonstrate a generally high level of knowledge about HIV and HIV testing among youth, but persistent barriers, including stigma and confidentiality concerns, impact the uptake of HIV testing.

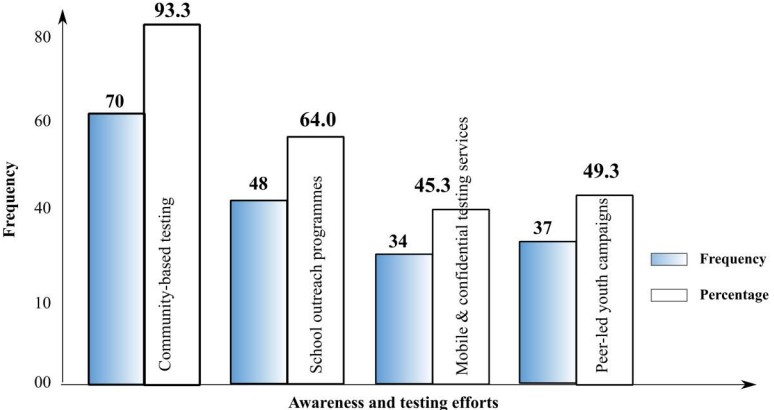

**Fig 3. Improvements to existing awareness and testing efforts.**

## Discussions

**Knowledge and awareness of HIV and testing services.** The study found that participants exhibited a high level of awareness about HIV, with 96% identifying sexual transmission as the primary mode of transmission. This finding aligns with previous research highlighting the importance of sexual transmission in HIV education [42]. However, awareness of other transmission routes, such as mother-to-child transmission and transmission through blood transfusion, was lower (46.7% and 30%, respectively), indicating a gap in comprehensive HIV education. This knowledge gap may be due to insufficient targeted education campaigns focused on these less prominent transmission routes in SSA [15,43,44].

Additionally, while 90.7% of participants were aware of HIV testing services, only 62.7% could accurately describe the testing procedure. This suggests that, although HIV testing services are known, there is a lack of detailed understanding about the testing process. Such a gap in knowledge could discourage individuals from seeking HIV testing, as misconceptions about the procedure may contribute to fear and reluctance as reported in Vietnam and Malawi [45,46]. The results emphasise the need for more comprehensive public health campaigns that not only raise awareness but also provide clarity on the specifics of HIV testing procedures.

**Experience with and attitudes toward HIV testing.** The findings revealed generally positive attitudes towards HIV testing, with 82.7% of participants indicating that they viewed knowing one's HIV status as the main benefit of testing. This suggests that youth in Kumba understand the personal health implications of HIV testing, which is crucial for public health interventions [13,47–49]. Furthermore, the high proportion of participants (93.3%) who reported having undergone HIV testing is encouraging, as it demonstrates an openness to testing and an understanding of its importance.

However, while the majority of participants expressed comfort with HIV testing, 17.3% reported feeling uncomfortable with the process. This discomfort was primarily driven by fear of a positive result and concerns about stigma. This finding is consistent with research by [50], who noted that stigma and fear of discrimination remain significant barriers to HIV testing, particularly in contexts where there is limited public education about HIV.

Interestingly, although most participants indicated they would encourage others to get tested (89.3%), a small but significant proportion (10.7%) would not, citing fear of stigma and breach of confidentiality. This illustrates the complex relationship between individual willingness to undergo testing and the community's stigma surrounding HIV, as noted in other African contexts [13,47,48,51]. This reluctance to encourage peers to get tested highlights the need for interventions that address social stigma, ensuring that HIV testing is seen as a routine and non-judgemental health behaviour.

**Perceptions of stigma and barriers to HIV testing.** Stigma and fear of social labelling were prominent themes in this study. Despite a generally positive attitude towards HIV testing, participants expressed concerns about how others might perceive them if they were to test for HIV. Female participants, in particular, noted that undergoing testing could be seen as a sign of promiscuity, while male participants mentioned fears related to masculinity and vulnerability. These gendered perceptions are well-documented in the literature, where HIV testing is often associated with negative stereotypes, especially for women and young men in sub-Saharan Africa [13,22]. The study highlights that social stigma continues to be a substantial barrier to HIV testing, even in communities where there is significant awareness and understanding of HIV.

Furthermore, concerns about confidentiality were also identified as barriers. Approximately 46.7% of participants indicated that they were worried about the confidentiality of HIV testing, particularly in public health settings. This is consistent with findings from other studies, which suggest that concerns about privacy can deter individuals from seeking HIV testing in sub-Saharan Africa, including Cameroon [43,52,53]. Confidentiality concerns are particularly pressing in small, close-knit communities where the fear of social judgement can have lasting consequences. The perceived lack of privacy during testing must be addressed to reduce these fears and encourage greater uptake of HIV testing.

**Community-based strategies for encouraging HIV testing.** Regarding community-based strategies, participants overwhelmingly supported the idea of improving access to HIV testing services through more community-based initiatives.

Almost all respondents (90.7%) were aware of existing HIV campaigns, and 86.7% had either participated in or observed these campaigns. This suggests that community-based campaigns are reaching a significant portion of the population. However, only 69.3% of participants found these campaigns to be highly effective. The discrepancy between awareness and perceived effectiveness could reflect concerns about the inclusivity and sensitivity of these campaigns [20,54]. Some participants indicated that campaigns did not always cater to the youth population or were not conducted in a manner that respected privacy.

Participants recommended several improvements to existing strategies, such as ensuring that testing services are more youth-friendly and that they provide greater emphasis on privacy and confidentiality. These recommendations are in line with the literature on effective HIV testing interventions, which emphasise the need for youth-oriented services that respect individuals' privacy and offer non-judgemental, confidential environments [20,55,56]. Additionally, the preference for mobile clinics and peer outreach underscores the importance of reaching youth where they are, rather than expecting them to seek out services in traditional healthcare settings, as reported by [57,58] in Nigeria and Bangkok. Community leaders and local health workers can play a crucial role in promoting these strategies and ensuring that they are accessible to all members of the community.

## Strengths and limitations of the study

**Strengths of the study.** One of the key strengths of this study is the use of FGDs to gather data from a diverse group of youth in Kumba. This method allowed for rich, qualitative data collection, providing insight into the complex factors influencing HIV testing uptake. The use of NVivo software to analyse the data also enhanced the rigour of the thematic analysis, ensuring that the findings were well-organised and meaningful.

**Limitations of the study.** Despite its strengths, this study has several limitations. First, the study was conducted in only four of the six targeted quarters of Kumba, which may limit the generalisability of the findings to the wider youth population. Second, the study employed a qualitative design with focus group discussions conducted between 17 February and 24 March 2025. While this approach provided in-depth insights into knowledge, attitudes, and barriers related to HIV testing, it represents a single point in time and cannot capture changes in behaviours or perceptions over time. Third, reliance on self-reported data could have introduced biases, as participants may have been inclined to provide socially desirable responses, particularly regarding their attitudes towards HIV testing.

Additionally, the data collectors, who were trained research assistants with experience in conducting qualitative interviews and community engagement, could have influenced responses through their interaction style or presence. Although efforts were made to minimise interviewer bias through standardised training, pre-testing of guides, and adherence to ethical procedures, their background and approach may have shaped how participants responded.

Future studies should aim to include a larger and more representative sample and consider combining qualitative methods with quantitative surveys to capture a broader range of perspectives.

## Conclusion

This study highlights the high level of awareness of HIV testing services among youth in Kumba, but it also reveals the persistent barriers of stigma and concerns about confidentiality. Despite these challenges, the results suggest that community-based strategies and youth-friendly interventions could significantly improve HIV testing uptake. To reduce stigma and enhance the effectiveness of HIV testing campaigns, it is crucial to address confidentiality concerns and ensure that testing services are non-judgemental and accessible to all youths.

## Recommendations

Based on the findings, the study recommends the following strategies to improve HIV testing uptake among youth in Kumba:

Increase the availability of youth-friendly, confidential HIV testing services, including mobile clinics and peer outreach programmes.

Strengthen community-based HIV awareness campaigns that focus on addressing stigma, particularly through targeted gender-sensitive approaches.

Enhance education about the different modes of HIV transmission and the testing procedure to address misconceptions and provide clarity on the testing process.

These recommendations, if implemented, could foster a more supportive environment for HIV testing and contribute to improving public health outcomes in the community.

## Supporting information

**S1 Appendix. FGD checklist.**
(PDF)

**S2 Appendix. FGD dataset minimal anonymised.**
(XLSX)

**S3 Appendix. COREQ checklist.**
(PDF)

## Acknowledgements

The authors wish to thank the participants of the study for their availability during data collection.

We wish to immensely acknowledge the HIV Free staff of Centre Médical d'Arrondissement (CMA) Ntam and Catholic Integrated Health Centre (CIHC) Kumba for inputs in fieldwork.

We also appreciate the contributions of the staff and students of the RAPHA Higher institute of health and biomedical science in Kumba.

## Author contributions

**Conceptualization:** Frederick Nchang Cho, Marie Clarie Fien Ndim, Christabel Afor Tatah.

**Data curation:** Frederick Nchang Cho, Marie Clarie Fien Ndim, Diane Zinkeng Tongwa.

**Formal analysis:** Frederick Nchang Cho, Marie Clarie Fien Ndim, Diane Zinkeng Tongwa.

**Investigation:** Marie Clarie Fien Ndim, Diane Zinkeng Tongwa.

**Methodology:** Frederick Nchang Cho, Marie Clarie Fien Ndim, Diane Zinkeng Tongwa, Andrew N Tassang, Franklin Ngwesse Ngome.

**Project administration:** Christabel Afor Tatah, Eugine Mbuh Nyanjoh.

**Resources:** Frederick Nchang Cho, Marie Clarie Fien Ndim, Eugine Mbuh Nyanjoh.

**Supervision:** Frederick Nchang Cho, Christabel Afor Tatah, Eugine Mbuh Nyanjoh, Andrew N Tassang, Franklin Ngwesse Ngome.

**Validation:** Frederick Nchang Cho, Marie Clarie Fien Ndim, Diane Zinkeng Tongwa, Christabel Afor Tatah, Eugine Mbuh Nyanjoh, Andrew N Tassang, Franklin Ngwesse Ngome.

**Visualization:** Frederick Nchang Cho, Marie Clarie Fien Ndim, Eugine Mbuh Nyanjoh.

**Writing – original draft:** Frederick Nchang Cho, Marie Clarie Fien Ndim, Diane Zinkeng Tongwa, Christabel Afor Tatah, Eugine Mbuh Nyanjoh, Andrew N Tassang.

**Writing – review & editing:** Frederick Nchang Cho, Marie Clarie Fien Ndim, Diane Zinkeng Tongwa, Christabel Afor Tatah, Eugine Mbuh Nyanjoh, Andrew N Tassang, Franklin Ngwesse Ngome.

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
