## [Decision Letter · Decision Letter 0]

15 Oct 2025

Dear Dr. Cho,

Thank you for submitting your manuscript to PLOS ONE. After careful consideration, we feel that it has merit but does not fully meet PLOS ONE’s publication criteria as it currently stands. Therefore, we invite you to submit a revised version of the manuscript that addresses the points raised during the review process.

**ACADEMIC EDITOR: **

Results: Include verbatim participant quotes to illustrate each theme., Clarify whether percentages represent coded response frequencies rather than statistical generalizations., Reformat Figure 2 and correct decimal errors.

References: Replace Wikipedia and other non-peer-reviewed sources with official or peer-reviewed references., Verify all references exist and are accessible; remove or update any future-dated citations.

Data Availability: Provide a Data Availability Statement that complies with PLOS ONE policy. Anonymized transcripts, coding framework, or minimal dataset should be deposited in a public repository (e.g., Dryad, Figshare). If full transcripts cannot be shared, provide a clear justification.

Ethics:  Add detail on how confidentiality was maintained during FGDs (e.g., anonymization, secure storage).

Presentation: Proofread for grammar, tense consistency, and formatting, Ensure all tables/figures are properly titled, numbered, and referenced, Upload the FGD guide/checklist as supplementary material.

We look forward to receiving your revised manuscript.

Kind regards,

Hamufare Dumisani Mugauri, Ph.D. Medicine and Health Sciences

Academic Editor

PLOS ONE

Additional Editor Comments (if provided):

Reviewers' comments:

Reviewer's Responses to Questions

**Comments to the Author**

1. Is the manuscript technically sound, and do the data support the conclusions?

Reviewer #1: Yes

Reviewer #2: Yes

2. Has the statistical analysis been performed appropriately and rigorously?

Reviewer #1: Yes

Reviewer #2: N/A

3. Have the authors made all data underlying the findings in their manuscript fully available?

Reviewer #1: Yes

Reviewer #2: No

4. Is the manuscript presented in an intelligible fashion and written in standard English?

Reviewer #1: Yes

Reviewer #2: Yes

Reviewer #1: 1- Findings: have to write separately with code examples from samples.

2- Discussion : Have to write separately in organized and structured manner.

3- Conclusion : have to write separately in structured manner.

Reviewer #2: I would like to acknowledge and sincerely appreciate the efforts of Mr. Frederick Nchang Cho, the corresponding and submitting author, and the entire research team for conducting this important study. Their work addresses a critical issue in understanding the knowledge, attitudes, and barriers to HIV testing among youth in Kumba, Cameroon, through a community-based approach. The study demonstrates thoughtful design and a strong commitment to improving practice and the quality of care.Line 36: Please include “Aim” as a subheading in the abstract to clearly present the study’s objective and improve readability.Results

Line 44: Please report the number of male and female participants to provide a clearer demographic overview.

**Do you want your identity to be public for this peer review?** For information about this choice, including consent withdrawal, please see our Privacy Policy

Reviewer #1: No

Reviewer #2: **Yes: ** Zainab Mahmood Dilshad Al-Zadjali

---

## [Author Response · Author response to Decision Letter 1]

24 Oct 2025

Response to Academic Editor and Reviewers

Dear Editor,

We thank the Academic Editor and Reviewers for their helpful and constructive feedback, which has greatly improved our manuscript. Below is a concise summary of our revisions.

Editor Comments

• Corrected study dates and clarified the sampling approach as multistage with purposive selection at the final stage.

• Added explanation of data saturation and confidentiality procedures.

• Included anonymised dataset and coding framework as supplementary materials, in line with PLOS ONE data policy.

• Added verbatim quotes to illustrate key themes and clarified that percentages represent coded response frequencies.

• Reformatted tables and figures, corrected captions and decimal errors, and ensured consistent referencing.

• Revised ethics statement, grammar, and presentation throughout.

Reviewer #1 Comments

• Revised the title to include study design and updated the abstract with clear objectives.

• Added participant demographics, defined key terms, and provided relevant WHO, UNAIDS, and national references.

• Clarified the research gap, sampling strategy, and definition of “active youth groups.”

• Expanded description of data collectors’ roles, focus group facilitation, and neighbourhood coverage.

• Confirmed descriptive statistics only and clarified that reliability tests such as Cronbach’s alpha do not apply to qualitative data.

• Retitled and reformatted tables for accuracy and improved readability.

• Expanded the limitations section to address timing, cross-sectional design, and fieldwork considerations.

General Updates

• Updated author details, ORCID IDs, and affiliations.

• Added COREQ checklist and anonymised dataset as supplementary files.

• Conducted full proofreading for clarity, tense consistency, and formatting.

We sincerely thank the Editor and Reviewers for their valuable guidance and for the opportunity to improve our paper.

Kind regards,

Frederick Nchang Cho (Corresponding Author)

Marie Clarie Fien Ndim (Co-Corresponding Author)

---

## [Editor Report · Decision Letter 1]

4 Nov 2025

Knowledge, Attitudes, and Barriers to HIV Testing Among Youth in Kumba, Cameroon: A Cross-Sectional Qualitative Community-Based Focus Group Study

PONE-D-25-23676R1

Dear Dr. Cho,

We’re pleased to inform you that your manuscript has been judged scientifically suitable for publication and will be formally accepted for publication once it meets all outstanding technical requirements.

Kind regards,

Hamufare Dumisani Mugauri, Ph.D. Medicine and Health Sciences

Academic Editor

PLOS ONE
---

## [Editor Report · Acceptance letter]

PONE-D-25-23676R1

PLOS ONE

Dear Dr. Cho,

I'm pleased to inform you that your manuscript has been deemed suitable for publication in PLOS ONE. Congratulations! Your manuscript is now being handed over to our production team.

Kind regards,

on behalf of

Dr Hamufare Dumisani Mugauri

Academic Editor

PLOS ONE